# Modification of Muscle-Related Hormones in Women with Obesity: Potential Impact on Bone Metabolism

**DOI:** 10.3390/jcm9041150

**Published:** 2020-04-17

**Authors:** Laurent Maïmoun, Thibault Mura, Vincent Attalin, Anne Marie Dupuy, Jean-Paul Cristol, Antoine Avignon, Denis Mariano-Goulart, Ariane Sultan

**Affiliations:** 1Département de Médecine Nucléaire, CHRU Montpellier, 34090 Montpellier, France; d-mariano_goulart@chu-montpellier.fr; 2Physiologie et Médecine Expérimentale du Cœur et des Muscles (PHYMEDEX), U1046 INSERM, UMR9214 CNRS, Université de Montpellier, 34295 Montpellier, France; a-avignon@chu-montpellier.fr (A.A.); a-sultan@chu-montpellier.fr (A.S.); 3Department of Biostatistics, Clinical Epidemiology, Public Health, and Innovation in Methodology, Nîmes University Hospital, University of Montpellier, 30029 Montpellier, France; Thibault.MURA@chu-nimes.fr; 4Département Endocrinologie, Nutrition, Diabète, Equipe Nutrition, Diabète, CHRU Montpellier, 34090 Montpellier, France; v-attalin@chu-montpellier.fr; 5Service de Biochimie, CHU de Montpellier, 34090 Montpellier, France; am-dupuy@chu-montpellier.fr (A.M.D.); jp-cristol@chu-montpellier.fr (J.-P.C.)

**Keywords:** myokines, obesity, areal bone mineral density, bone remodelling markers

## Abstract

Lean body mass (LBM) is a determinant of areal bone mineral density (aBMD) through its mechanical actions and quite possibly through its endocrine functions. The threefold aims of this study are: to determine the effects of obesity (OB) on aBMD and myokines; to examine the potential link between myokines and bone parameters; and to determine whether the effects of LBM on aBMD are mediated by myokines. aBMD and myokine levels were evaluated in relation to the body mass index (BMI) in 179 women. Compared with normal-weight controls (CON; *n* = 40), women with OB (*n* = 139) presented higher aBMD, myostatin and follistatin levels and lower irisin levels. Except for irisin levels, all differences between the OB and CON groups were accentuated with increasing BMI. For the whole population (*n* = 179), weight, BMI, fat mass (FM) and LBM were positively correlated with aBMD at all bone sites, while log irisin were negatively correlated. The proportion of the LBM effect on aBMD was partially mediated (from 14.8% to 29.8%), by log irisin, but not by follistatin or myosin. This study showed that myokine levels were greatly influenced by obesity. However, irisin excepted, myokines do not seem to mediate the effect of LBM on bone tissue.

## 1. Introduction

Several studies have demonstrated that subjects with obesity present higher areal bone mineral density (aBMD) than normal-weight subjects [1,2,3,4,5]. More recently, we confirmed the favourable effect of obesity on aBMD, highlighting that this phenomenon was not only accentuated with age and obesity severity, but also depended on sex, with bone tissue being more sensitive to obesity in women than men [6]. The increase in body mass that accentuates the mechanical loading on the skeleton is assumed to be the major factor in the bone mass gain in this population [1]. Indeed, bone is an adaptive tissue that has the capacity to modify its mass and microarchitecture in response to a mechanical stimulus [7]. This assumption may explain the accentuated aBMD gain with increases in the body mass index (BMI) at the hip, a weight-bearing bone site [8]. However, more surprisingly, a gain in aBMD and microarchitecture adaptations were also reported at the radius [3,4,6,9,10], a less mechanically solicited bone site, and these changes cannot be induced only by the gravitational forces associated with increased body weight. These results suggest that systemic biologically active molecules, related to obesity, may interact with bone metabolism. Intuitively, the first studies were oriented toward adipose tissue, which is principally impacted by obesity. Increased leptin production [11,12] and oestrogen synthesis by adipocytes [13,14] have been found to have positive actions on bone mass, thus limiting fracture risk [15]. However, in a large group of subjects with obesity, we confirmed [6] that aBMD correlates better with a dynamic load resulting from lean body mass (LBM) than a static load allocated to fat mass (FM) and that LBM is independently linked to aBMD [4,16,17,18]. These findings suggest that LBM, rather than FM, has a protective effect on bone tissue in this population. In addition to muscle force-generated mechanical signals [16,19], attention has recently been focused on muscle as an endocrine organ that regulates other biological targets, including pancreas, liver and adipose tissue [20]. Furthermore, these muscle-secreting factors, defined as “myokines”, biochemically affect bone metabolism in both endocrine and paracrine manners [21,22,23]. Among them, myostatin is involved in muscle mass and aBMD, as demonstrated in knockout mice [21,22,23], and in bone cell differentiation [24] and bone repair [25]. In humans, data regarding myokine effects on bone are relatively scarce, but a link between myostatin gene polymorphisms and peak bone mass acquisition has been demonstrated [26,27], as has the link with fracture risk [28]. In addition to the muscle-bone complex dialogue, myokines may be involved in the muscle-adipose complex by lipolysis and therefore have an impact on adipokine concentrations [29]. This, in turn, might have repercussions on bone formation and resorption.

Thus, the threefold aim of this study was to (1) determine the effects of obesity (OB) on areal bone mineral density (aBMD), bone turnover markers and specifically muscle-related hormone (myokines) levels; (2) examine the potential link between serum myokine levels and bone parameters; and (3) determine whether the effects of LBM on aBMD are mediated by myokines.

## 2. Subjects and Methods

Women with obesity defined by a BMI higher than 30 kg/m^2^ were consecutively recruited in the Nutrition Clinic of the University Hospital of Montpellier, France, between December 2012 and September 2016. All had been referred for metabolic and physical assessment of their obesity. Most of these patients had a longstanding history of obesity (more than 5 years) and none of the included patients had undergone bariatric surgery. Exclusion criteria were pregnancy, acute medical treatment and physical handicap (amputation, neurological lesion, orthopaedic prosthesis) that might interfere with body composition measurement. Moreover, patients with a body weight >190 kg or height ≥192.5 cm were also excluded due to limitations of the densitometry device. Medical history and menopausal status, when relevant, were obtained by questionnaires. Histories of smoking status and diabetes mellitus, as well as current medications, were recorded. Height and weight were measured wearing light clothing and no shoes; BMI was calculated as weight in kg divided by the square of height in metres (kg/m^2^). Waist circumference was recorded to the nearest 0.1 cm midway between the last rib and the crest of the ileum using a non-stretch tape measure. Each patient was categorised in a group according to the grade of obesity using the World Health Organisation (WHO) definition [30]: class I (BMI 30–34.9 kg/m²), class II (BMI 35–39.9 kg/m²) or class III (BMI ≥ 40 kg/m²). The control group was recruited by local advertisement or from friends of patients who agreed to participate. Individuals in the control group reported neither history of obesity (BMI ranging from 18 to 24.9 kg/m²), nor diabetes mellitus, hypertension or dyslipidaemia. Physical activity levels were not specifically determined. Nevertheless, the control group consisted of subjects who performed only leisure physical activities for fewer than one hour per week. None of the individuals with obesity were included in a training programme on the day of inclusion.

In order to have four groups of equivalent size and close in age, we randomly selected 5 to 7 women per 10-year age group from each BMI and control group (frequency matching) among 570 women with obesity and 120 controls initially recruited.

### 2.1. Patient Consent

All subjects gave written informed consent. The study was performed according to the principles of the Declaration of Helsinki and was approved by the local ethics committee (CPP Sud-Méditerranée IV, Montpellier, France, ethical approval number: N DC-2009–1052). All patients were entered into a registry with data collected during their hospitalisation, including anthropometric, clinical and biological information.

### 2.2. Bone Mineral Density, Body Fat and Lean Body Mass 

DXA (Hologic QDR-4500A, Hologic, Inc., Waltham, MA, USA) measured aBMD (g/cm^2^) of the whole body and at specific bone sites: anteroposterior lumbar spine (L1–L4), dominant arm radius and hip. The soft tissue body composition (fat mass (FM, kg), percentage of body fat mass (% FM) and lean body mass (LBM, kg)) were derived from the whole-body scan. All scanning and analyses were performed by the same operator to ensure consistency after following standard quality control procedures. Quality control for DXA was checked daily by scanning a lumbar spine phantom consisting of calcium hydroxyapatite embedded in a cube of thermoplastic resin (Hologic X-CALIBER Model DPA/QDR-1 anthropometric spine phantom, Hologic Inc., Waltham, MA, USA). The coefficient of variations (CVs) were 0.8% for spine and radius, 1.1% for the hip, and <1% for LBM and FM.

### 2.3. Assays 

Fasting blood samples (25 mL) were collected in the morning (08:30–10:00) in sterile chilled tubes by standard venipuncture technique. The samples were allowed to clot at room temperature and were then centrifuged at 2500 rpm for 10 min at 4 °C. Serum samples were stored at −80 °C until analysis. All samples were run in duplicate and, to reduce inter-assay variation, all the plasma samples were analysed in a single session.

Concerning bone metabolism, plasma samples were assayed by Cobas 6000© (Roche Diagnostic, Mannheim, Germany) for osteocalcin, and by iSYS© (IDS, Paris, France) for type I-C telopeptide breakdown products (sCTx) and tartrate-resistant acid phosphatase-5 (Trap-5). The inter- and intra-assay CVs for the latter three parameters were lower than 7%.

The determination of adipokines and myokines were performed by enzyme-linked immunosorbent assay (ELISA) assays. Adiponectin and leptin were measured with TECOmedical kits with intra-assay CV <5% and inter-assay CV from 7% to 10%). Lower limit of detection (manufacturer’s data) was <0.27 and 0.25 ng/mL, respectively. Myostatin was performed by Immunodiagnostic kit with intra- and inter-assay CV <10% and <13 %, respectively. Lower limit of detection (manufacturer’s data) was <0.37 ng/mL. Follistatin and Irisin were measured with Biovendor kits with intra- and inter- assay <5% and <8% for Follistatin and <8% and <10% for Irisin. Lower limit of detection (manufacturer’s data) was 0.03 ng/mL and <1 ng/mL, respectively.

## 3. Statistical Analysis

The patient characteristics are described with proportions for categorical variables and with means ± standard deviations (SD) for quantitative variables. Comparisons between patients with obesity and controls were performed using the Student *t*-test for quantitative variables when the data distribution was normal, and the Mann–Whitney U test when the variables were skewed. These variables were then compared between the three classes of obesity using analysis of variance (ANOVA) or the Kruskal–Wallis test. When the overall comparison was significant, we performed a two-by-two analysis with correction for multiple tests using the Bonferroni–Dunn method.

Correlations between anthropometric and biological parameters and aBMD were assessed using Pearson’s partial correlation coefficients adjusted for age. sCTx and irisin levels were not normally distributed and were logarithmically transformed to approach a linear relationship with aBMD.

In order to determine whether the effects of LBM on aBMD were mediated by myokines, we used the CAUSALMED procedure [31] in statistical analysis software (SAS) that uses linear regression adjustment methods [32] to estimate the percentage of the total effect of LBM on aBMD that can be attributed to the mediation by myokines. This percentage corresponds to the proportion to which the total effect of LBM on aBMD is reduced after controlling for the myokine level. The analyses of the three myokines (myostatin, follistatin and irisin) were performed separately and adjusted for height and weight.

Statistical analyses were performed at the conventional two-tailed α level of 0.05 using SAS version 9.4 (SAS Institute, Cary, NC, USA).

## 4. Results

The sample included 179 subjects, 139 women with obesity (OB) and 40 normal-weight women (controls, CON). Their anthropometric characteristics are summarised in Table 1. The age distribution ranged from 19.9 to 77.4 years, with a comparable mean age of 47.0 ± 15.2 years and 45.6 ± 16.9 years, respectively, for OB and CON. As expected, weight, BMI, LBM and FM (kg and %) were significantly higher in OB compared with CON (*p* < 0.001). No significant difference (*p* = 0.565) was observed for the mean age of menopause (CON: 51.5 ± 5.0, 49.4 ± 7.8 for class I, 49.7 ± 6.4 for class II and 48.9 ± 4.4 years for class III). Similarly, the use of tobacco according to present, never or previous status were comparable between groups (*p* = 0.772).

### 4.1. Areal Bone Mineral Density

Compared with CON, women with OB presented significantly higher aBMD values at all bone sites (whole body (−5.2%), total proximal femur (−13.9%), lumbar spine (−9.7%) and radius (−4.4%)), as well as high Z-score values (Table 1**)**. These differences were particularly marked at the hip, a weight-bearing bone site, in the women with class III obesity, as demonstrated by the highest *Z*-score.

### 4.2. Biological Parameters

Regarding bone turnover markers (Table 1 and Figure 1), women with OB presented significantly lower mean values not only for the bone formation marker (osteocalcin; *p* < 0.001), but also for the bone resorption markers (sCTx and Trap-5, *p* < 0.001) compared with CON. Moreover, the reduction in bone remodelling was accentuated with an increase in BMI.

Regarding adipokines (Table 1 and Figure 2), as expected, the adiponectin level was lower (*p* < 0.001), while the leptin level was higher (*p* < 0.001) in patients with obesity compared with controls. The alteration of the leptin level was particularly marked in class III obesity.

Regarding myokines (Table 1 and Figure 2), the irisin level was lower (*p* < 0.001), while follistatin and myostatin levels were higher (*p* < 0.05), in patients with obesity compared with controls. For follistatin and myostatin, differences with the reference values were accentuated in class III obesity, while for irisin the difference was accentuated in class I and II obesity.

### 4.3. Correlations between Anthropometric and Biological Parameters and aBMD

Table 2 summarises the correlation coefficients between the anthropometric and biological parameters and aBMD adjusted by age. When the whole population was studied (participants with or without obesity), weight, BMI, FM, LBM and leptin were positively correlated with aBMD at all bone sites, while osteocalcin (OC), log sCTx, Trap-5, adiponectin and log irisin were negatively correlated. These correlations were comparable within bone sites. No correlation was observed between myostatin or follistatin level and aBMD.

In addition, myostatin, follistatin and leptin were positively correlated with the anthropometric parameters (weight, BMI, FM and LBM), while OC, log sCTx, Trap-5, adiponectin and log irisin were negatively correlated.

Finally, the myostatin level was positively correlated with leptin. Log irisin levels were positively correlated with OC and Trap-5 and negatively with leptin. The follistatin level was negatively correlated with OC and log sCTx.

### 4.4. Mediation Effects of Myokines on aBMD

Table 3 displays the increase in aBMD (effect) at various bone sites for a 10-kg increase in LBM. These results showed a significant effect of LBM on aBMD whatever the bone site, ranging from 0.014 to 0.063 g/cm². Interestingly, the minimal effect appeared at the radius, a non-weight-bearing bone site, while the maximal effect appeared at the hip, a weight-bearing site. In addition, the proportion of the LBM effect on aBMD was partially mediated (ranging from 14.8% to 29.8%) by log irisin. This effect appeared to be more marked at the radius. However, neither myostatin nor follistatin seemed to mediate this effect.

## 5. Discussion

In this study, we investigated the effects of obesity on circulating myokine levels and their potential effects on bone metabolism in women with obesity with broad ranges of age and obesity severity. Our main results highlight specific profiles of aBMD, bone turnover and myokine levels in the women with obesity compared with normal-weight subjects. Moreover, we observed that the effect of LBM on bone tissue was partially mediated by specific myokines.

Numerous studies have investigated the effects of obesity on bone mass and reported higher aBMD in patients compared with normal-weight subjects [1,2,3,4,5,6,8]. In our study, we observed similar results, but the aBMD differences between patients and controls were exacerbated at the hip in a weight-, BMI-, FM- and LBM-dependent manner. At the radius, a non-weight-bearing bone site, aBMD was also higher in patients, with no difference according to BMI. This finding suggests that circulating factors related to obesity may act favourably on bone tissue, probably with a threshold effect. This possibility is important to note because understanding the interactions between bone and muscle based on the identification of endocrine factors may open new therapeutic approaches in the management of osteoporosis and sarcopenia.

Until now, various hormonal changes, such as increased leptin or oestrogen levels, have been proposed to explain in part the higher bone mass described in obese patients [11,12,13,14]. However, the strong positive correlations between LBM and aBMD at each bone site observed in this study and in our previous work [6] may also suggest a positive effect of muscle on bone tissue. Based on these observations, skeletal muscle may enhance bone mass through at least two modes of action: (1) biomechanical function due to the increase in mechanical stimulation generated by muscle contractions and (2) endocrine and paracrine functions characterised by the release of growth factors and cytokines [33].

### 5.1. Obesity and Myokines

In the present study, we focused for the first time on the potential impact of several myokines on bone tissue in patients with obesity. Myostatin and follistatin levels were slightly higher in women with obesity compared with normal-weight subjects. Moreover, we demonstrated that myostatin and follistatin were positively associated with parameters of adiposity and LBM, as previously reported [34]. Higher circulating myostatin levels or gene expression have been found in obese animals [35] and patients with morbid obesity [36,37]. Moreover, higher myostatin protein levels in muscle were also reported in high-fat diet-induced obese rats [38] and the myostatin gene was found to be upregulated in the skeletal muscle of obese mice [39]. Interestingly, in subjects with obesity, both muscle myostatin expression and circulating follistatin levels decreased after weight loss due to bariatric surgery, in parallel with the decrease in LBM and the increase in insulin sensitivity [34,37,40]. These findings suggested a potential role of these hormones in the regulation of energy homeostasis in skeletal muscle [34,37].

In contrast, irisin levels were dramatically decreased in women with obesity. These results agree with previous clinical data [41,42] showing that both muscle mRNA and circulating irisin levels were negatively associated with obesity [41,42]. Furthermore, decreased plasma irisin levels were also reported in patients with metabolic syndrome [43], but, the effect of obesity on irisin is still under debate, with studies showing discrepant results [44,45,46]. A recent meta-analysis underlined [47] the wide range of circulating irisin levels reported in both subjects with obesity and normal-weight controls. Indeed, numerous other factors, including ethnicity, gender, BMI and age, seemed to affect the irisin levels. It is also possible that this conflicting data might also result from the target epitopes chosen by the manufacturers [42,47], as suggested by Albrecht et al. [48].

The question that remains is: Could changes in these myokine concentrations explain the increase in aBMD at weight- and non-weight-bearing sites in women with obesity?

Experimental studies have provided some evidence of the effects of myokines on bone tissue. For example, myostatin antibody treatment improved not only muscle mass and function, but also aBMD and bone formation in 18-month-old orchidectomised mice [49], with a direct effect on proliferation and osteoprogenitor cell differentiation [24]. In addition, Tang et al. 2016 [50] confirmed the negative effect of myostatin on bone by demonstrating that the administration of polyclonal myostatin antibody prevented bone loss and microarchitecture deterioration due to diet-induced obesity in rats. Age-related declines in aBMD were reported to be attenuated in myostatin-deficient mice [51], although, conversely, no direct link was observed between circulating myostatin levels and aBMD in various human populations like postmenopausal and elderly women [52,53] or young women with anorexia nervosa [54]. In our study of women with a broad age range and with or without obesity, we observed no association between myostatin levels and aBMD. Nevertheless, although myostatin was not correlated with bone turnover markers, we cannot exclude the possibility that it contributed to the reduced bone formation, particularly in the women with class III obesity who presented concomitantly higher myostatin and lower osteocalcin levels. To our knowledge, only Zhang et al. [26] found that myostatin genetic polymorphisms play a role in attainment of peak BMD in young Chinese women. Further, Wu et al. [55] reported recently that serum myostatin levels were negatively correlated with aBMD in elderly Chinese women. 

Regarding irisin, integrin αV/β5 was recently identified as the irisin receptor on osteocytes [56]. Moreover, irisin treatment upregulated the expression of sclerostin in osteocyte-like cells (MLO-Y4) and increased serum sclerostin levels in a dose-dependent manner when injected every day for six days in mice [56]. Conversely, the total deletion in these mice of the fibronectin type III domain-containing protein 5 (FNDC5), the precursor of irisin, led to reduced circulating levels of RANKL and increased femoral trabecular volume fraction. These mice were resistant to ovariectomy-induced bone loss through inhibition of osteoclastic bone resorption and osteolytic osteocytes [56]. Thus, lower irisin levels might be anabolic for bone, while chronically high irisin levels could promote bone catabolism by increasing sclerostin levels. In agreement with this probable action of irisin on bone tissue, we found that women with obesity presented concomitantly lower irisin levels and higher aBMD, and these two parameters were significantly and negatively correlated at all bone sites. The probability that irisin affected bone tissue in our population was reinforced by the positive correlation between irisin and bone remodelling markers. Osella et al. [43] also reported a lower level of circulating irisin in patients with metabolism syndrome, whereas Lu et al. [45] observed higher irisin levels in patients with obesity (*n* = 20) compared with healthy adults (*n* = 20). Further, irisin levels were positively correlated with total aBMD, muscle and fat mass. A very recent meta-analysis suggested that circulating irisin levels are decreased in middle-aged and older participants with osteoporosis or those with a history of fractures [57]. Irisin was also positively correlated with aBMD measured at the femoral neck and lumbar spine [57].

Interestingly, our mediation approach showed that aBMD gain in patients with obesity is due to an increase in LBM. Moreover, among the three myokines measured, the proportion of this effect was only mediated by irisin, although the proportion varied according to the bone site. The main effect was found at the radius, a non-weight-bearing bone site. This result may be explained by a lower effect of circulating irisin in this region, probably masked by the increase in mechanical loading due to obesity. Nevertheless, the proportion of the effect mediated by irisin on aBMD was modest. It would be interesting to analyse the effect of other myokines and pro-inflammatory cytokines (TNF-α and IL-6) on aBMD [50].

### 5.2. Limitations 

This study presents some limitations, particularly its observational cross-sectional design and the single measurements of both aBMD and the biological parameters. However, these limitations are mitigated by the large population studied, the wide age range, the high degree of age-matching between participants with and without obesity and the extensive study of various myokines. Moreover, the subgroup analysis according to the BMI class provides important new data and may be considered as a strength of the study. In the future, longitudinal studies in subjects with obesity and decreasing weight may help to better understand the relationship between myokines and aBMD and even elucidate the relationship between osteoporosis and sarcopenia. Unfortunately, we do not have data about the muscle microenvironment to correlate with those obtained at the circulating level, but this limitation was inherent to the design of the study. In addition, we recognise that physical activity levels may influence myokines levels [58]. However, despite this parameter was not measured in this study, we assume that none of the individuals with obesity, none of the controls practised an intense training that probably limited the impact on myokine secretions. Finally, future studies should be specifically designed to evaluate the effect of other parameters susceptible to modify myokine levels, notably the effect of menopause as oestrogen levels were found to be positively correlated with irisin levels in healthy women [59] and as ovariectomy induces an increase of resting irisin levels in female rats [60]. Moreover, better characterisation of fat distribution (visceral vs subcutaneous fat) may be more informative than BMI, because these both components have different biological functions and may contribute differently to myokine synthesis [61].

## 6. Conclusions

In conclusion, this study confirmed that obesity is associated with an increase in aBMD and reduced bone turnover. Moreover, we reported that myokines were largely influenced by obesity, but in a specific manner. However, irisin excepted, myokines do not seem to mediate the effect of LBM on bone tissue. The adaptation of bone tissue to obesity may be defined as a mixed model including an equilibrium between the increase in mechanical loading due to the increase in body weight applied to the skeleton and the derived humoral factors released from muscle and adipose tissue that need to be identified. Future studies in men could be interested and would test the impact of gender on the effects of myokines on bone tissue in obesity.

## Figures and Tables

**Figure 1 jcm-09-01150-f001:**
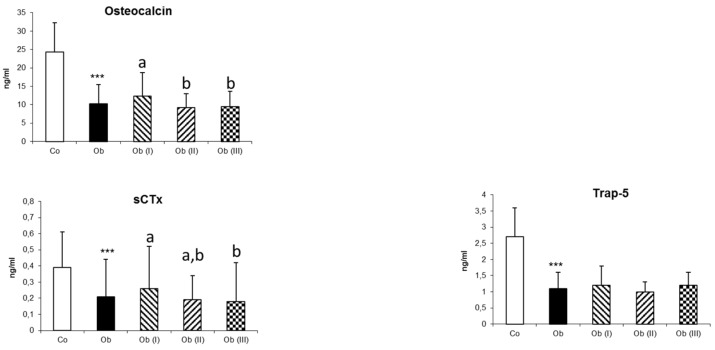
Markers of bone turnover in controls (CO) and in patients (Ob) according to the class of obesity. Data are presented as mean ± standard deviation. Patients with obesity were classified into three subgroups according to the body mass index (BMI) as follows: Class I obesity (Ob I): patients with BMI 30–34.9 kg/m²; Class II obesity (Ob II): patients with BMI 35–39.9 kg/m²; and Class III obesity (Ob III): patients with BMI ≥ 40 kg/m². *** indicates a significant difference between controls and patients with obesity for *p* < 0.001. For comparisons between subgroups of patients with obesity according to the BMI, there are no significant differences when two subgroups share the same letter (a or b) or no letter is displayed. sCTx: type I-C telopeptide breakdown product; Trap-5: Tartrate-resistant acid phosphatase-5.

**Figure 2 jcm-09-01150-f002:**
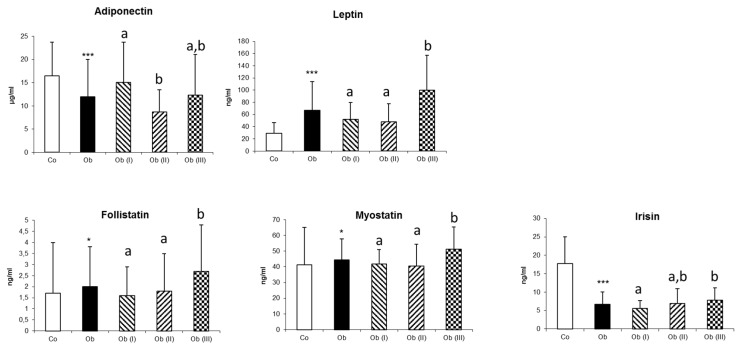
Adipokines and myokines in controls (CO) and in patients (Ob) according to the class of obesity. Data are presented as mean ± standard deviation. Patients with obesity were classified into three subgroups according to the body mass index (BMI) as follows: Class I obesity (Ob I): patients with BMI 30-34.9 kg/m²; Class II obesity (Ob II): patients with BMI 35-39.9 kg/m²; and Class III obesity (Ob III): patients with BMI ≥ 40 kg/m². * indicates a significant difference between controls and patients with obesity for *p* < 0.05 and *** for *p* < 0.001. For comparisons between subgroups of patients with obesity according to the BMI, there are no significant difference when two subgroups share the same letter or no letter (a or b) is displayed.

**Table 1 jcm-09-01150-t001:** Biological and anthropometric characteristics of patients, according to their obesity class, and controls.

Variables	Controls	Obese Patients	Class I Obesity	Class II Obesity	Class III Obesity	*p*-Values for Comparisons between Obese Groups
Number	40	139	47	45	47	
Age, years	45.6 ± 16.9	47.0 ±15.2	46.6 ± 15.7	46.6 ± 14.5	47.8 ± 15.5	0.912
Weight, kg	61.0 ± 7.8	101.2 ± 17.2 ***	87.5 ± 8.2 ^a^	97.1 ± 10.8 ^b^	118.9 ± 13.5 ^c^	<0.001
Height, cm	163.3 ± 6.0	162.1 ± 7.4	163.3 ± 7.2	161.0 ± 7.9	162.0 ± 7.2	0.344
BMI, kg/m²	22.8 ± 2.0	38.5 ± 5.9 ***	32.7 ± 1.4 ^a^	37.3 ± 1.4 ^b^	45.3 ± 4.3 ^c^	<0.001
**Body composition**						
Fat mass, kg	19.2 ± 5.1	46.4 ± 11.1 ***	36.8 ± 4.1 ^a^	44.6 ± 6.6 ^b^	57.9 ± 8.9 ^c^	<0.001
Fat mass, %	30.8 ± 5.4	45.0 ± 4.6 ***	41.5 ± 3.0 ^a^	45.1 ± 4.3 ^b^	48.3 ± 3.7 ^c^	<0.001
Lean body mass, kg	40.4 ± 4.9	53.7 ± 9.4 ***	49.7 ± 5.7 ^a^	51.0 ± 9.7 ^a^	58.2 ± 9.8 ^b^	<0.001
**Bone mineral density**						
Whole body (g/cm²)	1.056 ± 0.085	1.114 ± 0.100 ***	1.101 ± 0.103	1.101 ± 0.085	1.140 ± 0.106	0.093
Z-score whole body (SD)	−0.1 ± 0.9	0.8 ± 1.0 ***	0.7 ± 1.1	0.7 ± 0.9	1.1 ± 0.9	0.100
Hip (g/cm²)	0.886 ± 0.115	1.028 ± 0.130 ***	1.016 ± 0.142	1.004 ± 0.118	1.063 ± 0.123	0.069
Z-score hip (SD)	−0.1 ± 0.9	1.0 ±1.0 ***	0.9 ± 1.1 ^a,b^	0.8 ± 0.9 ^a^	1.4 ± 1.0 ^b^	0.022
Lumbar spine (g/cm²)	0.949 ± 0.132	1.051 ± 0.137 ***	1.048 ± 0.146	1.044 ± 0.134	1.061 ± 0.132	0.816
Z-score lumbar spine (SD)	−0.2 ± 1.1	0.9 ± 1.3 ***	0.9 ± 1.3	0.7 ± 1.3	0.9 ± 1.3	0.690
Radius (g/cm²)	0.558 ± 0.052	0.584 ± 0.048 **	0.581 ± 0.048	0.591 ± 0.048	0.579 ± 0.048	0.417
Z-score radius (SD)	0.9 ± 1.4	1.3 ± 1.0 *	1.3 ± 1.1	1.4 ± 1.0	1.2 ± 1.0	0.608
**Bone markers**						
Osteocalcin, ng/mL	24.3 ± 7.9	10.3 ± 5.1 ***	12.3 ± 6.4 ^a^	9.2 ± 3.8 ^b^	9.4 ± 4.2 ^b^	0.015
sCTx, ng/ml	0.39 ± 0.22	0.21 ± 0.23 ***	0.26 ± 0.26 ^a^	0.19 ± 0.15 ^a,b^	0.18 ±0.24 ^b^	0.006
Trap−5	2.7 ± 0.9	1.1 ± 0.5 ***	1.2 ± 0.6	1.0 ± 0.3	1.2 ± 0.4	0.067
**Adipokines**						
Adiponectin	16.5 ± 7.2	12.0 ± 8.0 ***	15.1 ± 8.6 ^a^	8.7 ± 4.8 ^b^	12.3 ± 8.8 ^a,b^	<0.001
Leptin	29.1 ± 17.4	67.1 ± 46.7 ***	52.1 ± 27.6 ^a^	48.3 ± 29.3 ^a^	100.1 ± 57.1 ^b^	<0.001
**Myokines**						
Myostatin	41.2 ± 23.8	44.5 ± 13.4 *	41.7 ± 9.4 ^a^	40.6 ± 13.8 ^a^	51.2 ± 14.2 ^b^	<0.001
Follistatin	1.7 ± 2.3	2.0 ± 1.8 *	1.6 ± 1.3 ^a^	1.8 ± 1.7 ^a^	2.7 ± 2.1 ^b^	0.007
Irisin	17.8 ± 7.2	6.7 ± 3.4 ***	5.6 ± 2.1 ^a^	6.9 ± 4.0 ^a,b^	7.8 ± 3.4 ^b^	0.004

Data are presented as mean ± standard (standard deviation, SD) deviation. Patients with obesity were classified into three subgroups according to the body mass index (BMI) as follows: Class I obesity: patients with BMI 30–34.9 kg/m²; Class II obesity: patients with BMI 35–39.9 kg/m²; and Class III obesity: patients with BMI ≥40 kg/m². * indicates a significant difference between controls and patients with obesity for *p* < 0.05, ** for *p* < 0.01 and *** for *p* < 0.001. For comparisons between subgroups of patients with obesity according to the BMI, there are no significant differences when two subgroups share the same letter (a, b, c) or no letter is displayed. BMI: body mass index; sCTx: type I-C telopeptide breakdown products; Trap-5: tartrate-resistant acid phosphatase-5.

**Table 2 jcm-09-01150-t002:** Pearson correlation coefficients of anthropometric parameters, bone turnover markers and myokines with areal bone mineral density adjusted for age.

	Weight	BMI	FM	LBM	aBMD WB	aBMD LS	aBMD Hip	aBMD Radius	Myostatin	Log Irisin	Follistatin
Weight	-	-	-	-	0.393 ***	0.345 ***	0.529 ***	0.282 **	-	-	-
BMI	-	-	-	-	0.345 ***	0.307 ***	0.483 ***	0.219 **	-	-	-
FM	-	-	-	-	0.299 ***	0.277 ***	0.450 ***	0.225 **	-	-	-
LBM	-	-	-	-	0.415 ***	0.359 ***	0.467 ***	0.316 ***	-	-	-
OC	−0.540 ***	−0.607 ***	−0.550 ***	−0.376 ***	−0.329 ***	−0.266 ***	−0.431 ***	−0.285 ***	−0.049	0.321 ***	−0.172 *
Log sCTx	−0.340 ***	−0.411 ***	−0.347 ***	−0.181 *	−0.296 **	−0.221 **	−0.356 ***	−0.207 **	−0.050	0.067	−0.207 *
Trap−5	−0.555 ***	−0.591 ***	−0.572 ***	−0.376 ***	−0.235 **	−0.269 ***	−0.387 ***	−0.260 ***	−0.080	0.424 ***	−0.101
Adiponectin	−0.240 **	−0.260 ***	−0.202 **	−0.211 **	−0.292 ***	−0.215 **	−0.266 ***	−0.162 *	−0.128	0.142	0.068
Leptin	0.524 ***	0.547 ***	0.573 ***	0.341 ***	0.198 *	0.208 **	0.198 *	0.155 *	0.248 **	−0.174 *	0.150
Myostatin	0.226 **	0.194 **	0.210 **	0.191 *	0.094	0.068	0.108	0.033	-	−0.011	0.008
Follistatin	0.162 *	0.192 *	0.174 *	0.053	0.037	−0.076	0.009	0.031	0.008	−0.004	-
Log Irisin	−0.449 ***	−0.447 ***	−0.458 ***	−0.345 ***	−0.309 ***	−0.281 ***	−0.368 ***	−0.297 ***	−0.011	-	−0.004

Data are presented as coefficients of correlation. * indicates a significant correlation for *p* < 0.05, ***p* < 0.01 and *** for *p* < 0.001. Abbreviations: BMI: body mass index; aBMD: areal bone mineral density: WB, whole body; LS: lumbar spine; FM: fat mass; LBM: lean body mass; OC: osteocalcin; sCTx: type I-C telopeptide breakdown products; Trap-5: tartrate-resistant acid phosphatase-5. sCTx and irisin levels were not normally distributed and were logarithmically transformed.

**Table 3 jcm-09-01150-t003:** Mediation effects of myokines on areal bone mineral density, adjusted for height and weight.

	WB aBMD			LS aBMD			Hip aBMD			Radius aBMD		
	Effect	SEM	*p*-Value	Effect	SEM	*p*-Value	Effect	SEM	*p*-Value	Effect	SEM	*p*-Value
Total effect for a 10-kg increase in LBM	0.039	0.007	<0.0001	0.049	0.012	<0.0001	0.063	0.010	<0.0001	0.014	0.004	0.0005
Proportion of the total effect of LBM mediated by	**%**	**STD**	***p*** **-Value**	**%**	**STD**	***p*** **-Value**	**%**	**STD**	***p*** **-Value**	**%**	**STD**	***p*** **-Value**
Myostatin	1.4	3.2	0.68	0.2	3.9	0.95	2.1	2.8	0.47	−0.1	5.9	0.98
Follistatin	−0.7	1.8	0.69	−2.1	3.1	0.49	−0.2	1.3	0.91	0.6	2.7	0.83
Log Irisin	14.8	7.6	0.05	17.8	9.3	0.06	15.8	6.3	0.01	29.8	15.2	0.05

aBMD: areal bone mineral density; LBM: lean body mass; WB: whole body; LS: lumbar spine. SEM: Standard Error of the Mean.

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
