# Peer review of "Modification of Muscle-Related Hormones in Women with Obesity: Potential Impact on Bone Metabolism"

_jcm, 2020, doi:10.3390/jcm9041150_

Round 1
Reviewer 1 Report
This is an interesting study investigating the association between obesity and myokines and their potential effect in bone metabolism among women. My comments are very minor and are listed below:
- How did the authors come up with the sample size? Had they done a power calculation? Is it just by chance that the 3 categories of people with obesity are balanced?
Why did they select only 30 healthy controls? - Results, page 4, line 161. The authors describe results on femoral neck BMD but I cannot see this outcome anywhere in the table
Author Response
Reviewer 1
Comments and Suggestions for Authors
This is an interesting study investigating the association between obesity and myokines and their potential effect in bone metabolism among women.
Dear reviewer thank you very much for these comments
My comments are very minor and are listed below:
- How did the authors come up with the sample size? Had they done a power calculation? Is it just by chance that the 3 categories of people with obesity are balanced?
Why did they select only 30 healthy controls? The number of healthy controls in our manuscript was n=40
Thank you for this pertinent comment. We are totally aware that the method used to constitute each group (controls vs. obese and obese groups according to BMI) lacked precision on this point. We have included in the revision manuscript in the paragraph "subject and methods (page 3, lines 99-101)" the missing informations and added the following sentence: “In order to have four groups of equivalent size and close in age, we randomly selected 5 to 7 women per 10-year age group from each BMI and control group (frequency matching) among 570 women with obesity and 120 controls initially recruited”.
More precisely, there were no data in the literature that would allowed us to calculate a sample size in a meaningful way. Then, we chose to select a sample size of more than 40 subjects per group for reasons of feasibility and because this sample size classically demonstrates medium to large effects sizes (Cohen, 1988). We initially had data on 570 obese women and 120 controls. In order to have four groups of equivalent size and close in age, we randomly selected 5 to 7 women per 10-year age group from each BMI group (frequency matching). The balance of the groups is not perfect due to the small number of individuals available for certain age groups in some BMI groups (this is why we have only 40 subjects in the control group). However, this method allowed us to obtain a comparable age between control and obese groups (45.6 ± 16.9 vs. 47.0 ±15.2, respectively) and between obese groups (p=0.912).
Reference: Cohen, J. (1988). Statistical power analysis for the behavioral sciences. Second Edition. Hillsdale, NJ: Lawrence Erlbaum Associates, Publishers.
- Results, page 4, line 161. The authors describe results on femoral neck BMD but I cannot see this outcome anywhere in the table
Thank you for this comment. The words “femoral neck [-15.1%]” was deleted. It was a mistake.
Reviewer 2 Report
The paper shows interesting data regarding the effects of muscle-related hormones, myokines, on bone metabolism and BMD, in overweight patients.
The work presents a clear and comprehensive revision of literature, as well as a detailed study comparing bone and muscle markers of overweight and normal-weight women, and their whole-body-DXA parameters.
I just suggest fery few spelling /typing errors, like at line 16: "corresponding" (c missing), at line 285 check the typing of "probability that irisin". Please, check again the whole text.
Author Response
Reviewer 2
The paper shows interesting data regarding the effects of muscle-related hormones, myokines, on bone metabolism and BMD, in overweight patients.
The work presents a clear and comprehensive revision of literature, as well as a detailed study comparing bone and muscle markers of overweight and normal-weight women, and their whole-body-DXA parameters.
Dear reviewer thank you very much for these comments
I just suggest fery few spelling /typing errors, like at line 16: "corresponding" (c missing), at line 285 check the typing of "probability that irisin". Please, check again the whole text.
All the typing errors had been corrected.